# Epidemic spread simulation in an area with a high-density crowd using a SEIR-based model

**Jibiao Zhou**[1,2], **Sheng Dong**[3], **Changxi Ma**[4]*, **Yao Wu**[5], **Xiao Qiu**[6]

**1** School of Civil and Transportation Engineering, Ningbo University of Technology, Ningbo, China,
**2** Department of Traffic Engineering & Key Laboratory of Road and Traffic Engineering, Ministry of Education, Tongji University, Shanghai, China, **3** School of Civil and Transportation Engineering, Ningbo University of Technology, Ningbo, China, **4** School of Traffic and Transportation, Lanzhou Jiaotong University, Anning District, Lanzhou, China, **5** School of Modern Posts & Institute of Modern Posts, Nanjing University of Posts and Telecommunications, Nanjing, China, **6** School of Civil Engineering and Transportation, Hohai University, Nanjing, China

\* machangxi@mail.lzjtu.cn

**Data Availability Statement:** The data used to support the findings of this study are included within the article.

**Funding:** This study was supported by Philosophy and Social Science Program of Ningbo (No. G20-ZX37), National Natural Science Foundation of

## Abstract

Understanding the spread of infectious diseases is an extremely essential step to preventing them. Thus, correct modeling and simulation approaches are critical for elucidating the transmission of infectious diseases and improving the control of epidemics. The primary objective of this study is to simulate the spread of communicable diseases in an urban rail transit station. Data were collected by a field investigation in the city of Ningbo, China. A SEIR-based model was developed to simulate the spread of infectious diseases in Tianyi station, considering four groups of passengers (susceptible, exposed, infected, and recovered) and a 14-day incubation period. Based on the historical data of infectious diseases, the parameters of the SEIR infectious disease model were clarified, and a sensitivity analysis of the parameters was also performed. The results showed that the contact rate (CR), infectivity (I), and average illness duration (AID) were positively correlated with the number of infections. It was also found that the length of the average incubation time (AIT) was positively correlated with the number of exposed individuals and negatively correlated with the number of infectors. These simulation results provide support for the validity and reliability of using the SEIR model in studies of the spread of epidemics and facilitate the development of effective measures to prevent and control an epidemic.

## Introduction

Infectious diseases are a type of disease caused by various pathogens that can be directly or indirectly transmitted from person to another, one animal to another or between people and animals. Epidemics of infectious diseases are occurring more often and spreading faster and further than ever in many different regions of the world [1]. Generally, infectious diseases are constantly evolving, emerging, and re-emerging, and an infection that is considered a national or global threat one year could be eliminated the next [2,3]. Initially, new infectious diseases could spread only as fast and far as the hosts could travel under their own power; however, in the context of globalization, the growth of trade activity, tourism and human migration is leading to increasingly widespread and rapid movement of disease vectors and, consequently, of the diseases they carry [4,5].

China (Grant No. 52002282), the Natural Science
Foundation of Zhejiang Province (No.
LQ19E080003, LY21E080010), and Philosophy
and Social Science Foundation of Zhejiang
Province (No. 21NDJC163YB).

**Competing interests:** The authors have declared
that no competing interests exist.

Subway stations [6], which constitute one of the busiest sites worldwide, involve passenger
connections and transfers. Due to the high concentration of passengers in subway stations, the
transmission of a virus is more widespread than in other public places [7,8], which poses
potential threats to the health and safety of the passengers. On the one hand, with the development
of urban rail transit networks, passenger flow continues to increase. For example, in
2019, a total of 40 cities in mainland China opened 6,720.27 kilometers of urban rail transit
operating lines, with a total of 968.77 kilometers of new operating lines. The total passenger
traffic volume reached 21.07 billion in 2018 [9], a year-on-year increase of 14%, with a total of
13.32 billion passenger trips and a total passenger turnover of 176.08 billion passenger-km
(pkm). On the other hand, infectious diseases [10] have plagued humans throughout history
and have even shaped history on some occasions. The plagues of biblical times, the Black
Death of the Middle Ages, and the "Spanish flu" pandemic of 1918 are a few examples [11].
Therefore, it is very important to study the spread of infectious diseases in public places, especially
in subway stations, and to determine effective prevention and control measures for infectious
diseases in public places.

## Literature review

The spread of infectious diseases has seriously affected public health and has posed severe challenges
to the global public health system. An epidemic of an infectious disease must have three
basic elements, namely, an infectious agent, transmission routes and susceptibility. The standard
public health emergency measures are usually the most efficient, including isolating the sources of
infection, cutting off or interrupting transmission routes, and providing special care for the most
susceptible people. For example, since the coronavirus disease 2019 (COVID-19) epidemic started
in Wuhan in late December 2019 [12,13], the Chinese government has taken robust measures to
curb the spread of the deadly virus, most notably ordering the full quarantine of Wuhan, the epicenter
of the outbreak, and implementing strong control and preventive measures in metropolitan
areas such as Beijing and Shanghai as well as other population centers around China.

At present, infectious disease prevention and control strategies mainly include [14–16]: (a)
immunization; (b) the development of effective treatments and vaccines and other medical prevention
and control measures; (c) isolation measures; (d) routine and practical epidemic prevention
measures (such as temperature detection, the use of protective masks, ventilation, daily
disinfection and epidemic prevention education, etc.); (e) school suspensions; (f) traffic control
(blocking traffic, prohibiting traffic) and other nonmedical prevention and control measures.
The above measures have played positive roles in preventing the spread of infectious diseases.

## The transmission of a virus by transport

Taking the COVID-19 epidemic as an example, there are three main transmission routes [17],
direct transmission, aerosol transmission, and contact transmission. When a host infected
with the virus coughs or sneezes, the virus sprays into the air along with saliva, mucus or other
body fluids. When the liquid is deposited onto a healthy potential host or the healthy individual
touches the surface of an object contaminated by these droplets, that individual may
become infected. The virus responsible for COVID-19, severe acute respiratory syndrome
coronavirus 2 (SARS-CoV-2), can also spread through the air in the form of aerosols, but aerosol
transmission is not the main route for this virus [18–21]. It is generally believed that interpersonal
interactions while using public transport in large cities may contribute to the spread
of influenza because public transport, such as subways or airplanes, involves relatively enclosed
spaces and dense traffic flow [22,23]. Previous studies have investigated the transmission
routes of epidemic viruses, which are briefly reviewed in this section.

About subway travel, Cooley et al. [24] conducted a simulation study to develop a unique influenza agent-based transmission model for New York City that explicitly represents subway riders as a transmission conduit, and they found that only 4% of transmissions occurred on the subway. The findings suggested that interventions targeted at subway riders would be relatively ineffective at containing the epidemic.

Regarding school, Lee et al. [25] conducted another simulation study to explore the effects of various school closure strategies on mitigating influenza epidemics with different reproductive rates (R0). The results indicated that any type of school closure may need to be maintained throughout most of the epidemic (i.e., at least 8 weeks).

Air travel, which involves more than three billion passengers annually, serves as a conduit for the spread of infectious diseases, including emerging infections and pandemics. Computational fluid dynamics (CFD) was used by Boulbene, et a. (2012) [26] to simulate the movement of air on a plane, and he found that when a passenger sneezes on the plane, the air flow actually helps spread the pathogen to other passengers. In a subsequent study, Weiss H. et al. (2019) [27] collected 229 environmental samples on ten transcontinental US flights and performed 16S rRNA sequencing. They found that although the microbiomes in airplane cabins had large flight-to-flight variations, they resembled the microbiomes of many other built environments.

Therefore, based on the above previous studies [16,27–29], if using public transit is unavoidable during an epidemic, moving around the vehicle should be avoided to minimize the possibility of infection.

## Modeling of infectious disease dynamics

The spread of infectious diseases can be unpredictable; fortunately, modeling techniques can help compensate for imperfect information gathered from large populations under difficult prevailing circumstances [30]. The infectious disease dynamics (IDD) approach, a mathematical technique, has developed into a rich interdisciplinary field. It is driven both by the desire for fundamental understanding and the need to use that understanding to aid public health decision making.

Compared with traditional statistical models, the IDD model can not only describe the process of disease development and transmission and predict the state of disease occurrence but also evaluate the effects of various prevention and control measures and provide decision-making support regarding the measures need to prevent and control diseases. At present, IDD models mainly include four types: compartment models (CMs), meta-population models (MMs), individual-based models (IMs), and network models (NMs).

The main idea underlying the CM model is to divide the population into several compartments, which, respectively represent agents in different disease states, and then dynamic equations of related variables are established by mathematical techniques. Finally, the dynamic process of disease transmission can be modeled. For example, Kermack and McKendrick (1991) [31,32] predicted the number and distribution of cases of an infectious disease as it was transmitted through a population over time and proposed the classic SR model. The SR model is a compartmental differential-equation model that structures the infected population in terms of the age of infection while using simple compartments for people who are susceptible (S) and recovered/removed (R). Unfortunately, the generalizability of this model is difficult to analyze, and a number of open questions remain regarding its dynamics.

In a subsequent study, the metapopulation model (MM) was proposed based on the classic IR model. The modeling concept underlying the metapopulation model is to simulate the migration behavior of individuals between populations, and an SIR model or SIS model can be used to simulate the infectious disease transmission process within the population. For

example, Watts D. et al. (2005) [33] modeled the movement of individuals between contexts via simple transport parameters and allowed diseases to spread stochastically using MMs. They found that when epidemics occur, the basic reproduction number $R0$ may bear little relation to their final size. Next, Colizza V. et al. (2007) [34] constructed a theoretical and computational microscopic framework for the study of a wide range of realistic MMs and agent-based models that include the complex features of real-world networks. The results provided a general theoretical understanding of the behavior of more realistic MMs.

The individual-based model (IM) is a microsimulation model that mainly includes a cellular automata model and an agent-based model. The modeling concept underlying the IM model is that individuals are treated as cells or agents with a limited set of state and behavior rules. By defining various rules of behavior, such as individuals' responses to the etiology of the disease, individuals' movement in space, and interactions among individuals, the final evolutionary behavior of a complex infectious disease system composed of the etiology, host and environment is eventually simulated. For instance, Milne G. et al. (2008) [35] simulated the transmission of and effectiveness of interventions for epidemic influenza in a community with an IM. The results indicated that multiple social distancing measures applied early and continuously could be effective at interrupting the transmission of the pandemic virus for R0 values up to 2.5.

Moreover, for the network model (NM), the main underlying modeling concept is to treat individuals in the population as nodes in a network, and the contact relationships between individuals are described by the edges between nodes in the network. For instance, Ajelli M. et al. (2010) [36] conducted a side-by-side comparison of the results obtained with a stochastic agent-based model and a structured, stochastic MM for the progression of a baseline pandemic event in Italy. The results indicated that both models yielded epidemic patterns that were in very good agreement at the level of granularity accessible by both approaches. Riley [37] reviewed the application of four methods (patch model, distance-transmission model, multi-group model, and NM) to four diseases (measles, foot-and-mouth disease, pandemic influenza, and smallpox). The results showed that household demographics have an important impact on the spatial transmission of human diseases, such as smallpox, influenza, and other infectious diseases.

## Simulation method

Given the characteristics of the spread of infectious diseases based on passenger flow through a rail transit station, passengers in a rail transit station and the SEIR model were chosen as the research subjects and the modeling method, respectively, for this study. Through the simulation of the spread of an infectious disease through the Tianyi station at Ningbo Rail Transit Line 1, the factors affecting the spread of the infectious disease, that is, the contact rate, the transmission ability, and the duration of the infectious disease, were quantitatively analyzed. In the context of rail transit, the speed of the spread of the disease and the infection rate can be affected by multiple factors. The most immediate factors include the number of infectious persons and their distribution among the passengers, the transmission route and the transmissibility of the infectious disease, and the level of immunity. Based on the five routes of transmission, namely, contact transmission, aerosol transmission, water and food transmission, insect transmission and others, this manuscript dissects the external and internal factors and the passenger-related factors that affect the spread of infectious diseases through rail transit systems. The external factors include the above transmission routes, while the internal factors include the temperature, humidity, facility layout, subway stations, air, hygienic conditions, density, and intensity of passengers in rail transit stations. The passenger-related factors include their basic characteristics, physical quality, behavioral habits, immunity, and medical history.

## The SEIR model

The SIS or SIR model is generally used in studies of the spread of infectious diseases. Based on the SI model, the SIS model can be used to simulate infectious diseases such as common influenza by adding the measurable characteristics of the disease, and considering whether individuals are susceptible or infected. To simulate the production of antibodies after recover and thus the acquisition of immunity to the disease, the SIR model was developed by introducing a third status, recovered. Patients do not exhibit symptoms for a certain period as they move from the susceptible status to the infected status, during which time they do not spread the disease. To simulate the status of exposed but asymptomatic, the status exposed was introduced, and the SEIR model was developed. The SEIR differs from the SIR models in that it adds the duration of the disease and is more suitable for infectious diseases with no infectious potential during the incubation period.

**(1) The basic assumptions.** To reflect the actual route selection behavior of rail transit passengers as much as possible and improve the reliability and effectiveness of the simulation results, the following assumptions were made:

i. Passengers choose the shortest routes to purchase tickets and enter and exit the station; all passengers are familiar with the process of entering and exiting the station and can handle basic business transactions independently.

ii. Infected individuals are not infectious during the incubation period of an infectious disease.

iii. The number of passengers who meet any given passenger is basically the same, which is expressed by the average contact rate.

The distributions of age, gender, health status and daily behavior of passengers in the rail transit station were investigated through a questionnaire survey. These data were combined with previous findings regarding relevant pedestrian behavior characteristics. The incubation period (exposure period) of the infectious disease was assumed to be 1–14 days, the recovered period was 30–60 days, and the transmissibility was 0.01–0.31.

The disease spreads through the contact message "infection". Thus, each passenger has four potential statuses: susceptible, exposed, infected, and recovered. Passengers are initially susceptible. If they are exposed to the pathogen, they will enter the exposed status, which means that they display symptoms and produce antibodies against the pathogen. After a certain period, they will become susceptible again.

**(2) SEIR model.** Considering the given incubation period, the SEIR model was used to model passengers with four statuses: susceptible (S), exposed (E), infected (I), and recovered (R). Each variable represents the number of passengers in the corresponding group. The probability of being infected $\beta$ represents the rate at which people move from $S$ and $I$ to $R$. Assuming that the total number of regions $N$ remains unchanged and that $N = S + E + I + R$ is satisfied, the differential equations are constructed as:

$$\begin{cases} \dfrac{dS}{dt} = \mu(N - S) - \dfrac{\beta SI}{N} - vS \\[2mm] \dfrac{dE}{dt} = \dfrac{\beta SI}{N} - \mu E - \sigma E \\[2mm] \dfrac{dI}{dt} = \sigma E - \gamma I - \mu I \\[2mm] \dfrac{dR}{dt} = \gamma I - \mu R + vS \end{cases} \quad (1)$$

where $v$ is the perturbation factor, which indicates the ratio from the susceptible state $S$ to the

**Table 1. Descriptive statistics for demographic information.**

| Age group | Gender | Shoulder width (m) |
|---|---|---|
| Younger (<30) | female | 0.380 |
| | male | 0.410 |
| Middle-aged (30–60) | female | 0.395 |
| | male | 0.419 |
| Older (>60) | female | 0.390 |
| | male | 0.405 |

recovered state $R$. Its physical meaning is the protective efficacy of an individual after vaccination, and a larger value indicates that government interventions are more effective, and vice versa. Next, $\mu$ is the natural mortality rate, $\beta$ is the infection rate, $\gamma$ is the recovery rate, and $\sigma$ is the rate of moving from exposure to infection.

**(3) The parameters of the SEIR.** There is an important relationship between the validation of model parameters and the authenticity and effectiveness of the simulation results. The parameters include the time scale, agent static parameters, agent walking speed and characteristic parameters of disease spread. The specific values are as follows:

*i*. The static parameters of the agent: the space occupied by the agent on the ground is called the passenger space. In the process of movement, the dynamics of the passenger space change in complex ways, and their specific sizes are difficult to measure. Therefore, only the static passenger space was considered in this study, which included the horizontal space and vertical space. The former mainly consists of the shoulder width and the safety buffer, while the latter consists of the stride and the safety space. Many studies have reported human body measurements stratified by gender and age, as shown in **Table 1**. The other data used in this study are from relevant studies and experience.

*ii*. The agent walking speed. In the simulation, the walking speed has a significant impact on the time to leave the station because age and gender affect walking speed [13,14]. Therefore, the following data were specifically set with reference to the statistical data published by Chinese and non-Chinese researchers on the walking speed of pedestrians, combined with the average walking speed of passengers at the Tianyi station measured with the preliminary survey, as shown in **Table 2**.

The structural characteristics of the rail transit station were mainly obtained from the indicator map and the field investigation to provide data to support the simulation.

*iii*. Spread parameters of the infectious disease. The spread parameters of the infectious disease reflect the process by which the virus is transmitted throughout the passenger flow and include the total population, infectivity, contact rate and average illness duration. Based on historical data on the spread of infectious diseases, a model of an infectious disease at Tianyi station was constructed. The total number of passengers was set at 1000. Since There are 4 entrances and exits at Tingyi Station, and 4 entrances and exits are open under normal circumstances. During the survey time, we took 15 minutes as a time interval to count the flow of pedestrians entering the station during working days and non-working days respectively, and finally summarized them in units of 1 hour. Hence, for the convenience of simulation statistics, the number of simulated people per unit hour was set to 1000.

**Table 2. Walking speed stratified by age and gender.**

| Characteristic | Young females | Young males | Middle-aged females | Middle-aged males | Older adults & juveniles of both genders |
|---|---|---|---|---|---|
| Average speed (m/s) | 1.45 | 1.51 | 1.39 | 1.47 | 1.00 |

### The calibration of the parameters

The SEIR model uses differential equations to reflect the relationship versus time of the number of individuals in each of the four different statuses, namely, S, E, I and R, which has certain practical value for the study of the spread of infectious diseases. The static parameters and the walking speed of agents do not change with research objectives and environments; therefore, they can be regarded as constants and do not need to be calibrated.

The contact rate, infectivity coefficient, duration of exposure, and duration of infectiousness introduced are more abstract, which increases the difficulty of the practical application of the model. The accuracy of these parameters is the key to model construction and the correctness of the conclusion, but they differ across infectious diseases and transmission environments. Therefore, optimizing these parameters was an important part of this work.

Calibration experiments [38–40] were established for the SEIR model, and a table of the historical data was also established based on the collected epidemic data. The total number of passengers in the model was set at 1000, and it was assumed that there was an infectious source in the target station. The historical data was used as the objective function for multiple iterations, and the best results were saved at the end of the iterations.

When the total number of passengers was 1000, the result of the 476th iteration for the number of infected passengers had the best fit among all 501 iterations. When the amount of data is small, the result has a better fit. Through the model calibration process [41,42], the calibrated parameter values were obtained, as shown in **Table 3**.

### Simulation results

After the construction of the infectious disease model, the simulation experiments were carried out and repeated several times to obtain different datasets. According to the literature [33–35], the main parameters that affect epidemic spread include the number of initial patients, the contact probability, the transmissibility of the infectious disease, the duration of the infectious disease and the time of infection. The influence of other factors on the spread of the infectious disease due to the contact probability, transmissibility and duration of the infectious disease based on the experimental results was analyzed quantitatively by the control variable method of adjusting the values of the model parameters.

### The impact of the transmission ability of the infectious disease

The contact rate (CR) was set to 8.781, that is, the average number of persons contacted per minute, the duration of infectiousness was 53.112, and the duration of exposure was 6.332. The simulations were carried out by adjusting the transmissibility of the infectious disease. The results are shown in **Fig 1**.

**Table 3. Parameter values after experimental calibration.**

| Parameter | Description | Value | Unit | Data source |
|---|---|---|---|---|
| $N$ | total population | 1,000 | person | Average hourly cross-sectional headcount for field research |
| $t$ | time | 10–20 | $d$ | 1s is used in the simulation to represent 1d |
| $v$ | perturbation factor | 79.34% | — | Effectiveness of protection after individual vaccination |
| $\mu$ | natural mortality rate | $2*10^{-5}$ | person/unit time | Global natural mortality, as of March 2021 |
| $\beta$ | infection rate | $6.93*10^{-9}$ | person/unit time | MCMC parameter estimation |
| $\gamma$ | recovery rate | 0.12 | person/unit time | MCMC parameter estimation |
| $\sigma$ | the rate of moving from exposure to infection | 1/5.7 | person/unit time | In the actual epidemic, the inverse of the incubation period |

Note: MCMC for Markov Chain Monte Carlo.

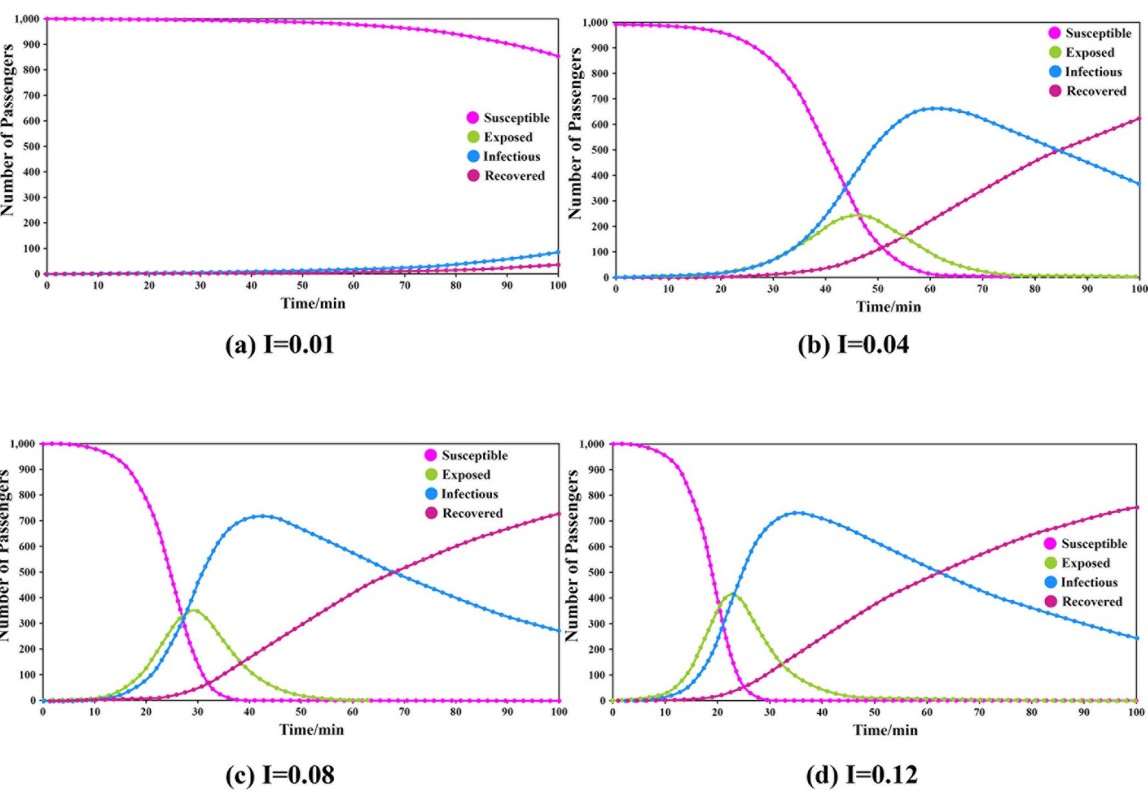

**Fig 1. Results of different groups under different conditions of transmissibility.**

The results show that the transmissibility was lower when I = 0.01, and the number of passengers in different statuses changed slightly. When I = 0.04, the peak number of exposed passengers were 245, which appeared at the 46th minute, and the peak number of infected passengers was 669, which appeared at the 61st minute. When I = 0.08, the corresponding values were 352 exposed at the 29th minute and 717 infected at the 42nd minute. When I = 0.12, the corresponding values were 415 exposed at the 22nd minute and 730 infected at the 35th minute.

## The impact of contact rate on the spread of the infectious disease

The transmissibility was set to 0.110, the duration of infectiousness was 53.112, and the duration of exposure was 6.332. The simulations were carried out by adjusting the CR. The results are shown in **Fig 2**.

The results show that when CR = 3.0, the peak number of exposed passengers was 236, which appeared at the 48th minute, and the peak number of infected passengers was 662, which appeared at the 64th minute. When CR = 7.0, the corresponding values were 366 exposed at the 27th minute and 720 infected at the 40th minute. When CR = 12.1, the corresponding values were 450 exposed at the 19th minute and 735 infected at the 32nd minute. When CR = 15.0, the corresponding values were 486 exposed at the 17th minute and 738 infected at the 30th minute.

## The impact of the duration of infectiousness on the spread of the infectious disease

The transmissibility was set to 0.110, the CR was 8.781, and the duration of exposure was 6.332. The simulations were carried out by adjusting the duration of infectiousness. The results are shown in **Fig 3**.

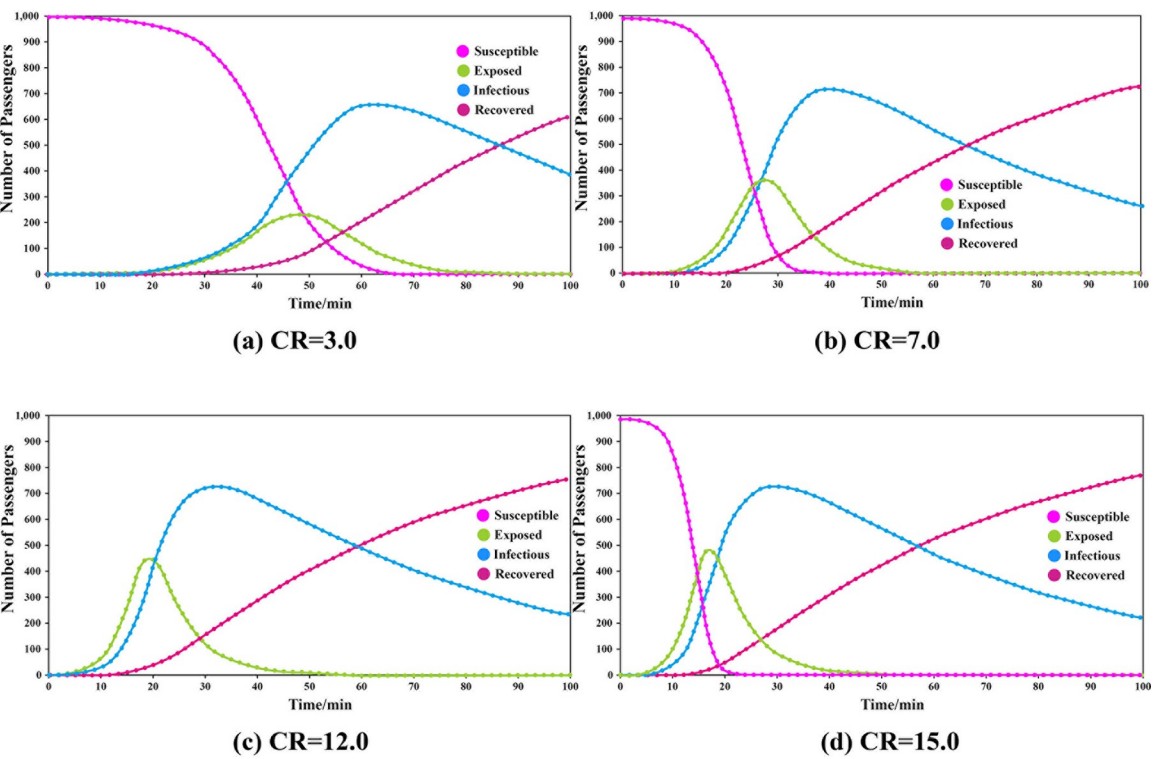

**Fig 2. Results of different groups under different contact rates.**

The results show that when AID = 3.0, the peak number of exposed passengers was 199, which appeared at the 43rd minute and the peak number of infected passengers was 91, which appeared at the 46th minute. When AID = 24.0, the corresponding values were 387 exposed at the 25th minute and 576 infected at the 34th minute. When AID = 45.0, the corresponding values were 400 exposed at the 24th minute and 700 infected at the 45th minute. When AID = 60.0, the corresponding values were 403 exposed at the 24th minute and 747 infected at the 60th minute.

## The impact of the duration of exposure on the spread of the infectious disease

The transmissibility was set to 0.110, the CR was 8.781, and the duration of infectiousness was 53.112. The simulations were carried out by adjusting the duration of exposure. The results are shown in **Fig 4**.

The results show that when AIT = 1.0, the peak number of exposed passengers was 155, which appeared at the 12th minute, and the peak number of infected passengers was 887, which appeared at the 18th minute. When AIT = 6.4, the corresponding values were 404 exposed at the 24th minute and 726 infected at the 37th minute. When AIT = 10.5, the corresponding values were 475 exposed at the 29th minute and 652 infected at the 46th minute. When AIT = 14.0, the corresponding values were 515 exposed at the 33rd minute and 604 infected at the 53rd minute.

## Sensitivity analysis

A sensitivity analysis is a quantitative analysis of the degree to which the changes in the values of several factors affect one or more key indicators. The function of a sensitivity analysis to

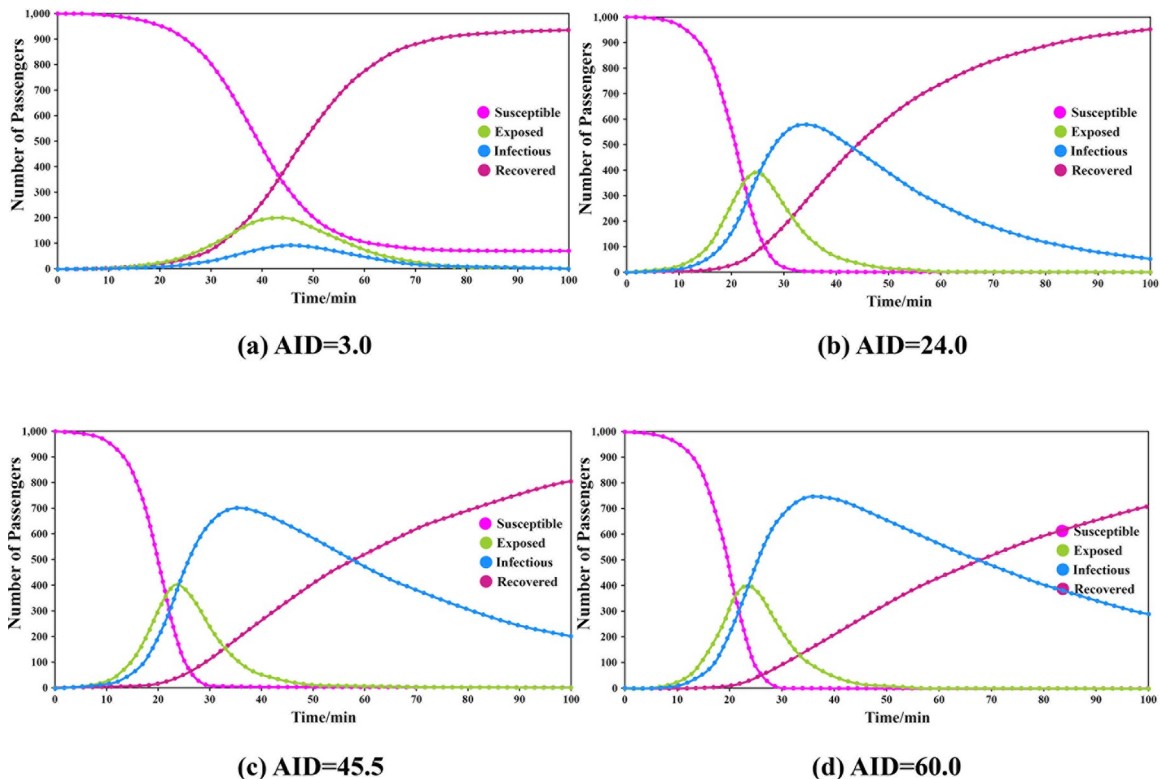

**Fig 3. Results of different groups under different durations of infection.**

explain the law governing the influence of these factors on the key indicators by changing the values of the relevant variables one by one. To further discuss the influence of the four parameters, transmissibility, CR, duration of exposure, and duration of infectiousness, on the numbers of exposed and infected individuals, sensitivity analyses were performed using the control variables method.

**(1) The transmissibility.** The simulation experiments were carried out with the preliminary calibrated data, constraining the range of infectivity. The simulation step was set to 0.01, and then the curves for the infection duration and the exposure duration were output. The results are shown in **Table 4**.

After 31 iterations, it can be seen in **Fig 5** that when $0.01 \leq I \leq 0.31$, the stronger the transmissibility was, the larger the peak numbers of exposed and infected individuals, and the shorter it took to reach those peaks. When I = 0.31, the peak number of exposed individuals were 743, which appeared at the 26th minute, and the peak number of infected individuals was 557, which appeared at the 13th minute.

**(2) Parameter CR.** The simulation experiments were carried out with the preliminary calibrated data, constraining the range of the CR. The simulation step was set to 0.01, and then the curves for the infection duration and the exposure duration were output. The results are shown in **Table 5**.

After 29 iterations, it can be seen in **Fig 6** that when $1 \leq CR \leq 15$, the larger the CR was, the larger the peak numbers of exposed and infected individuals, and the shorter the time needed to reach those peaks. When CR = 15, the peak number of exposed individuals was 738, which appeared at the 30th minute, and the peak number of infected individuals was 486, which appeared at the 17th minute.

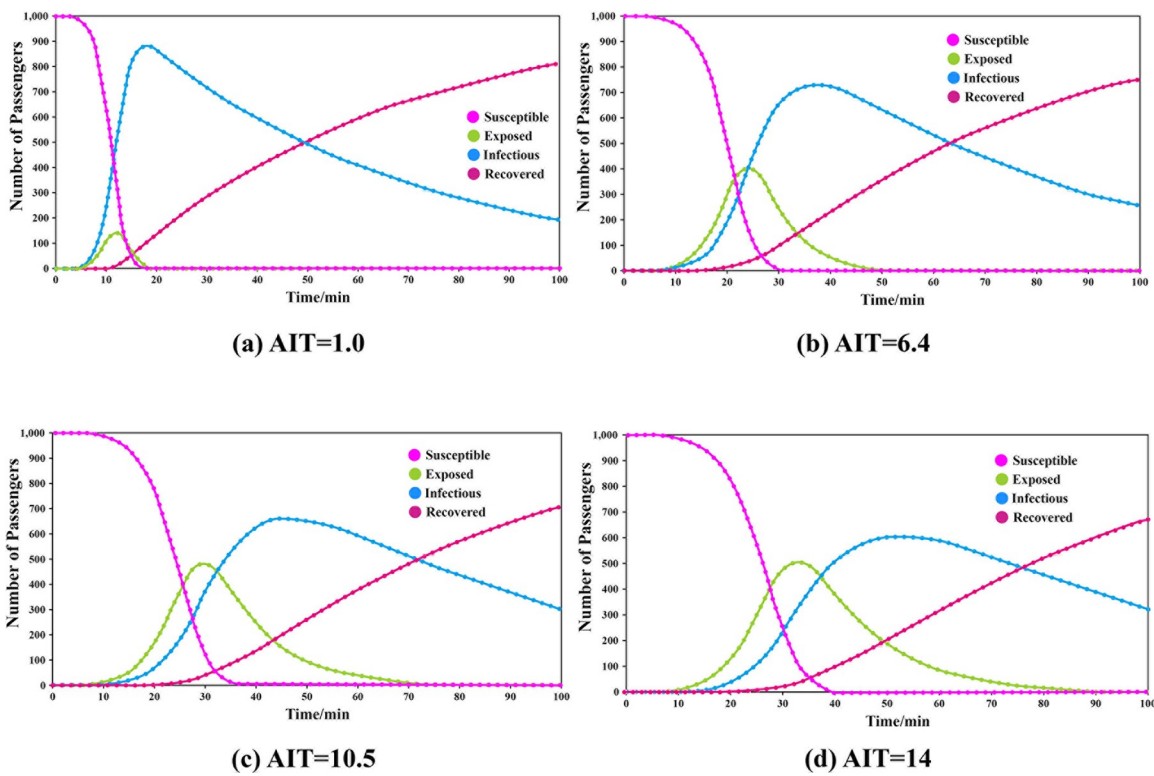

**Fig 4. Results of different groups under different durations of exposure.**

**(3) The duration of infectiousness.** The simulation experiments were carried out with the preliminary calibrated data, constraining the range of the duration of infectiousness. The simulation step was set to 0.01, and then the curves for the infection duration and the exposure duration were output. The results are shown in **Table 6**.

After 58 iterations, it can be seen in **Fig 7** that when 3≤AID≤60, the longer the duration of infectiousness was, the larger the peak numbers of exposed and infected individuals. However, the duration of infectiousness did not obviously affect the time to reach the peaks. When AID = 60, the peak number of exposed passengers was 747, which appeared at the 37th minute, and the peak number of infected passengers was 403, which appeared at the 24th minute.

**(4) The duration of exposure.** The simulation experiments were carried out with the preliminary calibrated data, constraining the range of the duration of exposure. The simulation step was set to 0.01, and then the curves for the infection duration and the exposure duration were output. The results are shown in **Table 7**.

After 27 iterations, it can be seen in **Fig 8** that when 1≤AIT≤14, the longer the duration of exposure was, the larger the peak numbers of exposed individuals, the smaller the peak

**Table 4. Parameter values in the sensitivity analyses (I).**

| No. | Parameters | Values | Steps |
|---|---|---|---|
| 1 | Total Population | 1000 | - |
| 2 | Infectivity (I) | 0.01–0.31 | 0.01 |
| 3 | Contact Rate (CR) | 8.781 | - |
| 4 | Average Illness Duration (AID) | 53.112 | - |
| 5 | Average Incubation Time (AIT) | 6.332 | - |

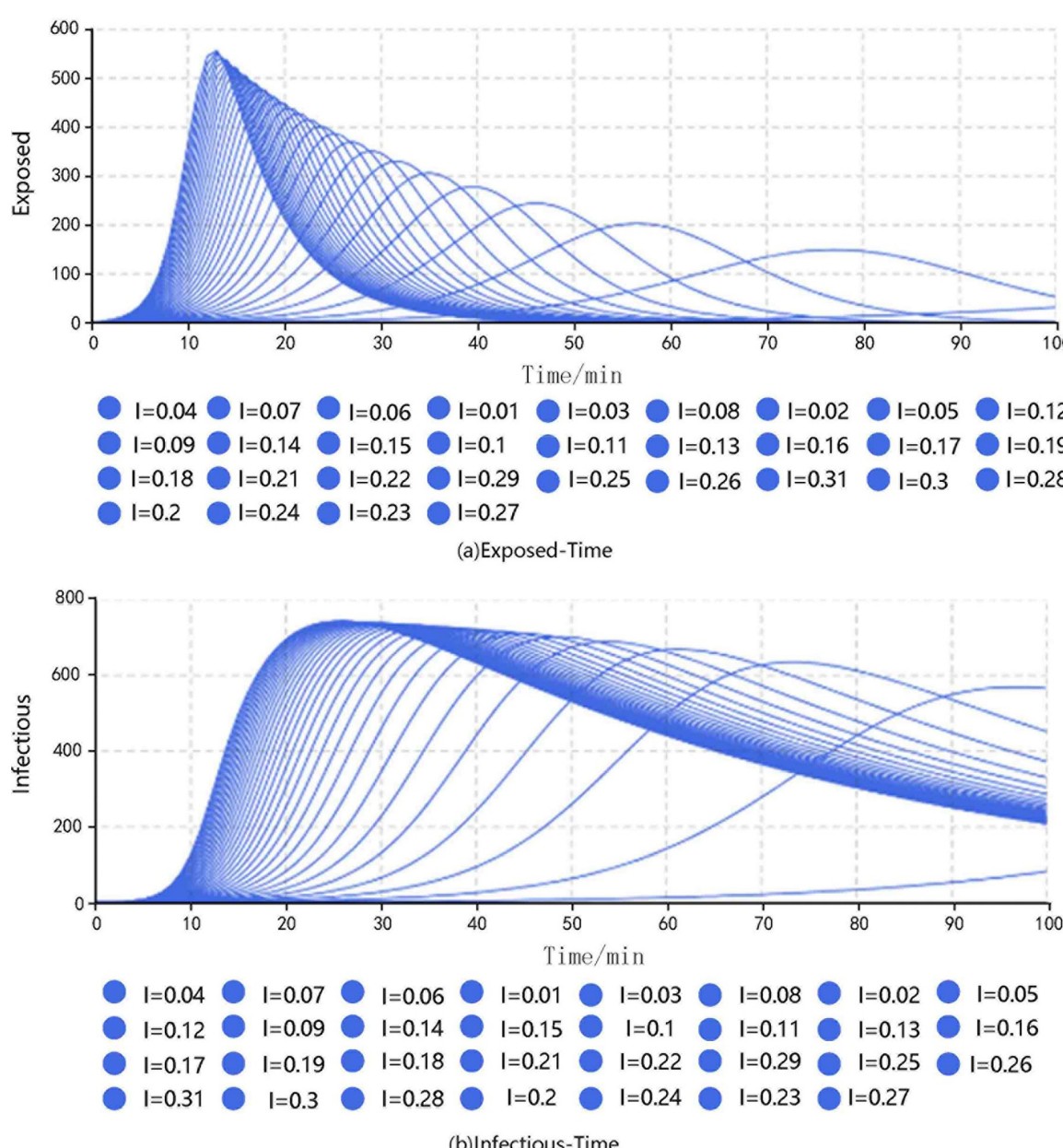

**Fig 5. Exposure-infection curve under different transmissibility conditions.**

Table 5. Parameter values in the sensitivity analyses (CR).

| Number | Parameters | Values | Steps |
|--------|------------|--------|-------|
| 1 | Total Population | 1000 | - |
| 2 | Infectivity (I) | 0.110 | - |
| 3 | Contact Rate (CR) | 1–15 | 0.5 |
| 4 | Average Illness Duration (AID) | 53.112 | - |
| 5 | Average Incubation Time (AIT) | 6.332 | - |

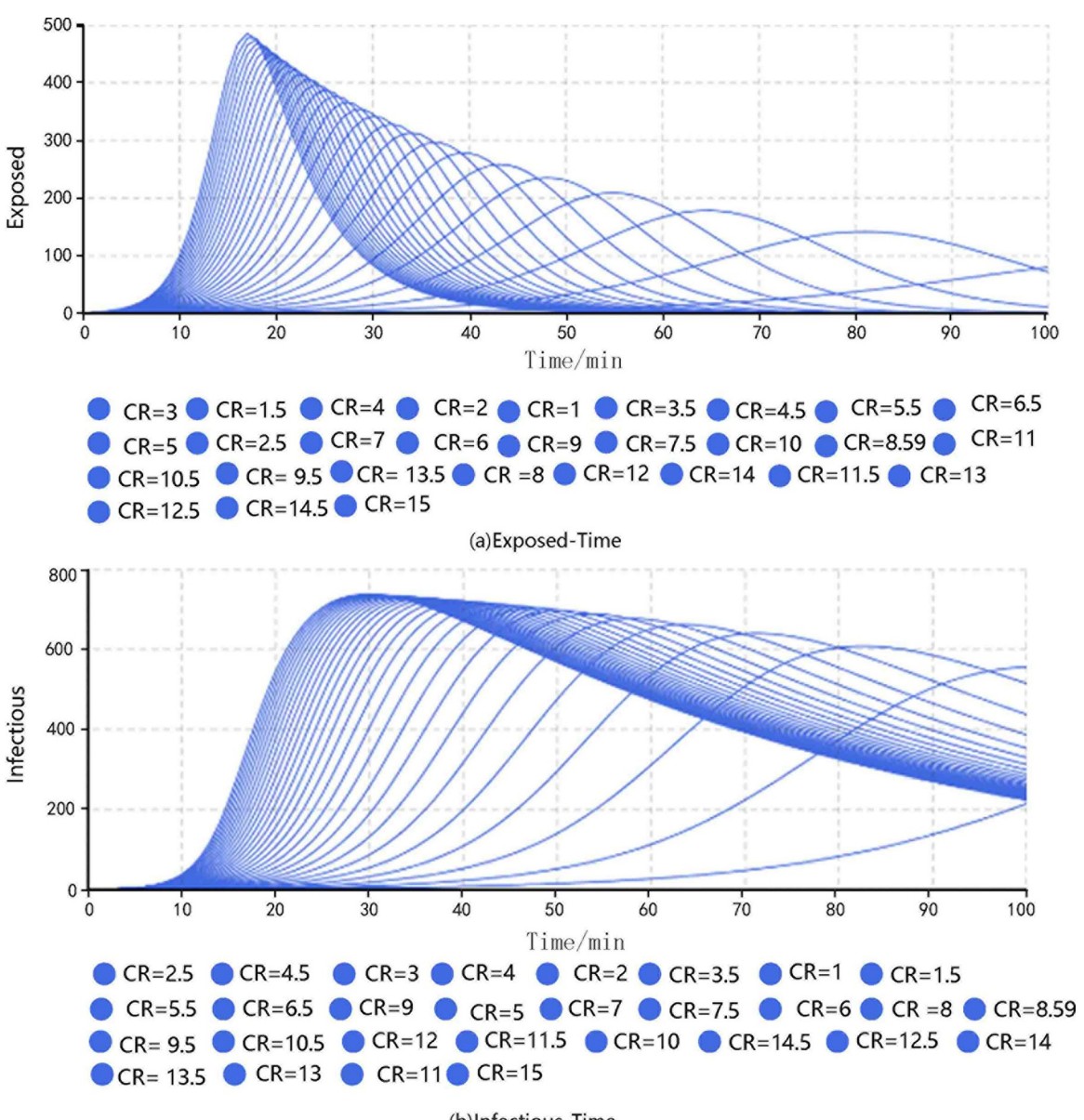

**Fig 6. Exposure-infection curve under different contact rates.**

**Table 6. Parameter values in the sensitivity analyses (AID).**

| Number | Parameters | Values | Steps |
|--------|-----------|--------|-------|
| 1 | Total Population | 1000 | - |
| 2 | Infectivity (I) | 0.110 | - |
| 3 | Contact Rate (CR) | 8.781 | - |
| 4 | Average Illness Duration (AID) | 3–60 | 1 |
| 5 | Average Incubation Time (AIT) | 6.332 | - |

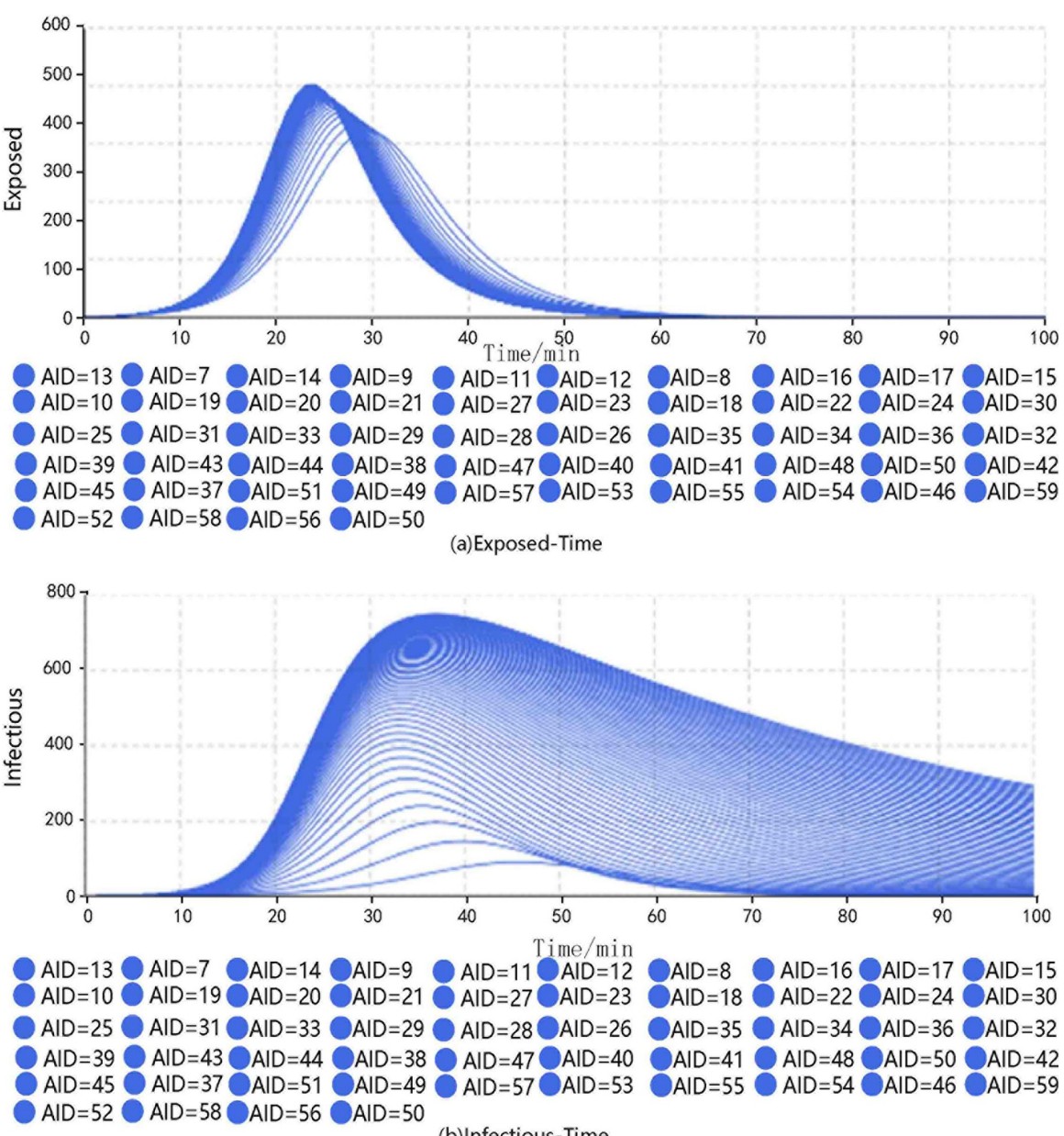

**Fig 7. Exposure-infection curve under different durations of infection.**

**Table 7. Parameter values in sensitivity analyses (AIT).**

| Number | Parameters | Values | Steps |
|--------|------------|--------|-------|
| 1 | Total Population | 1000 | - |
| 2 | Infectivity (I) | 0.110 | - |
| 3 | Contact Rate (CR) | 8.781 | - |
| 4 | Average Illness Duration (AID) | 53.112 | - |
| 5 | Average Incubation Time (AIT) | 1–14 | 0.5 |

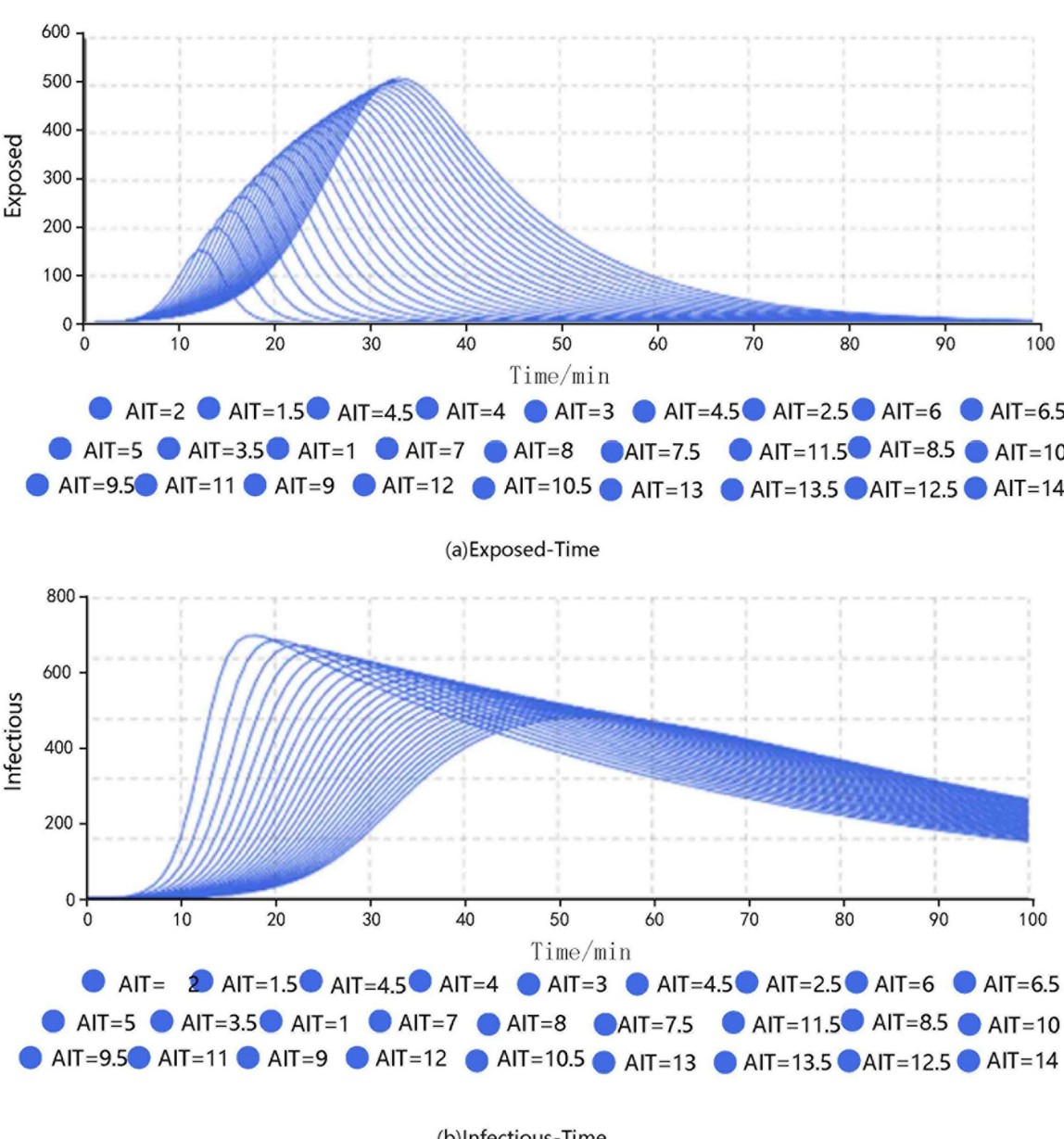

**Fig 8. Exposure-infection curve under different durations of exposure.**

number of infected individuals, and the longer they needed to reach those peaks. When AIT = 14, the peak number of exposed individuals was 604, which appeared at the 53rd minute, and the peak number of infected individuals was 516, which appeared at the 33rd minute.

Therefore, when I = 0.31, CR = 15, AID = 60, and AIT = 1, the peak number of infected individuals was 928, which appeared at the 8th min, as shown in **Fig 9**.

## 5. Conclusion and future study

In this study, the Tianyi rail transit station of Ningbo Rail Transit Line 1 was chosen as an example and used to construct an SIR model of the spread of an infectious respiratory disease in rail transit stations. Then, simulation experiments of the spread of the infectious disease

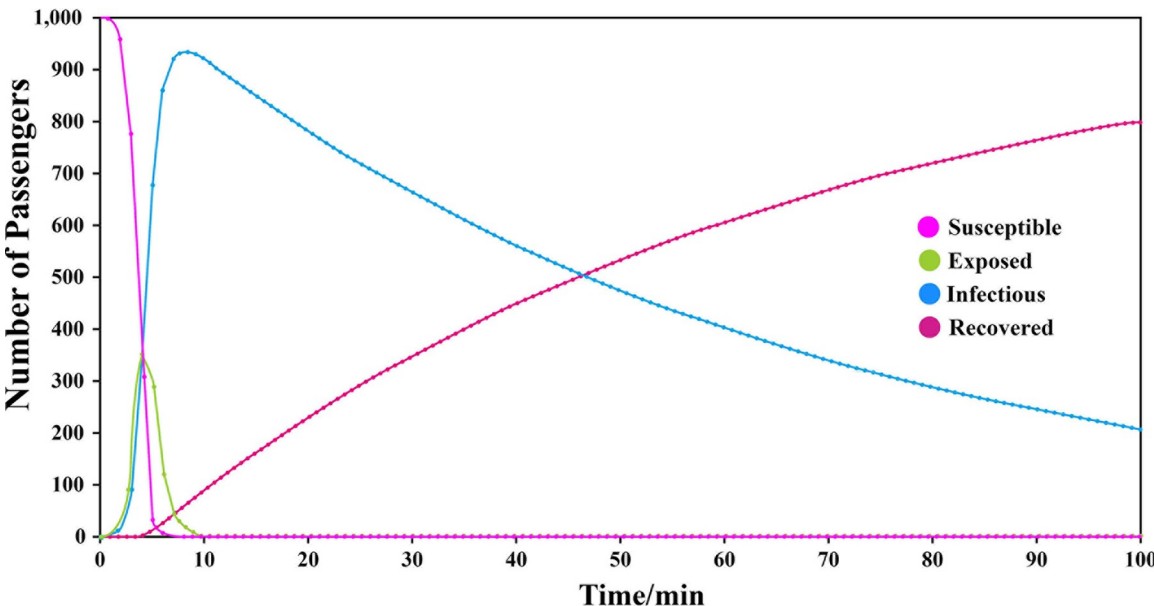

**Fig 9. The population dynamics of different groups.**

through the passenger flow in the rail transit station were designed and carried out. The simulation results show that the CR, transmissibility, duration of infectiousness and duration of exposure have important impacts on the spread of infectious diseases.

1. With increases in the CR and the transmissibility, the peak numbers of infected and exposed passengers also increased, with a shorter time needed to reach those peaks.

2. The longer the duration of infectiousness was, the larger the peak numbers of infected and exposed passengers, but there was little impact on the time needed to reach those peaks.

3. The longer the duration of exposure was, the larger the peak number of exposed individuals, the smaller of the peak number of infected individuals, and the longer the time needed to reach those peaks.

Therefore, the peak number of affected individuals in rail transit stations can be reduced by decreasing the CR, transmissibility, duration of infectiousness and duration of exposure.

The process and mechanism of the spread of an infectious disease through high-density passenger flow in a rail transit station were simulated and analyzed through the construction of this model, which provides theoretical support for the application of similar models in research on controlling the spread of infectious diseases. In addition, the research results can be used as a reference to help rail transit stations and disease control centers thoroughly understand the spread of infectious diseases in rail transit stations and formulate and implement effective disease control and prevention measures in other public places.

In this study, the process by which infectious diseases spread in rail transit stations were discussed preliminarily. There were still issues need further study, such as changing the facilities and organizational mode in the station and verifying the effectiveness of prevention and control measures. The model setting does not consider the impact of the temperature and ventilation in the station on the spread of the infectious disease. In the future, the model will be improved, and many practical factors will be considered to provide evidence that can be used to inform the development of effective measures to prevent the spread of infectious diseases.

## Acknowledgments

The authors would like to thank the undergraduates at the School of Civil and Transportation Engineering at Ningbo University of Technology for their assistance in field data collection, and double thank Haoyang Meng, Ruichao Qi, and Ziqiang Li for their great help in graphic polishing. The authors also appreciate the anonymous reviewers for their constructive comments and valuable suggestions to improve the quality of the manuscript.

## Author Contributions

**Conceptualization:** Jibiao Zhou, Sheng Dong, Changxi Ma, Xiao Qiu.

**Data curation:** Sheng Dong, Yao Wu, Xiao Qiu.

**Formal analysis:** Yao Wu, Xiao Qiu.

**Funding acquisition:** Jibiao Zhou, Sheng Dong.

**Investigation:** Sheng Dong, Yao Wu, Xiao Qiu.

**Methodology:** Jibiao Zhou, Sheng Dong.

**Project administration:** Jibiao Zhou, Changxi Ma.

**Resources:** Changxi Ma.

**Software:** Xiao Qiu.

**Supervision:** Changxi Ma.

**Validation:** Jibiao Zhou.

**Writing – original draft:** Jibiao Zhou, Sheng Dong, Xiao Qiu.

**Writing – review & editing:** Jibiao Zhou, Changxi Ma.

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
