## [Decision Letter · Decision Letter 0]

26 Mar 2021

PONE-D-21-06411

Epidemic Spread Simulation in an Area with a High-density Crowd Using a SEIR-Based Model

PLOS ONE

Dear Dr. Ma,

Thank you for submitting your manuscript to PLOS ONE. After careful consideration, we feel that it has merit but does not fully meet PLOS ONE’s publication criteria as it currently stands. Therefore, we invite you to submit a revised version of the manuscript that addresses the points raised during the review process.

We look forward to receiving your revised manuscript.

Kind regards,

Yanyong Guo, Ph.D

Academic Editor

PLOS ONE

Journal Requirements:

2. Please note that in order to use the direct billing option the corresponding author must be affiliated with the chosen institute. Please either amend your manuscript to change the affiliation or corresponding author, or email us at plosone@plos.org with a request to remove this option.

3. We note you have included a table to which you do not refer in the text of your manuscript. Please ensure that you refer to Tables 4-7 in your text; if accepted, production will need this reference to link the reader to the Table.

Reviewers' comments:

Reviewer's Responses to Questions

**Comments to the Author**

1. Is the manuscript technically sound, and do the data support the conclusions?

Reviewer #1: Yes

Reviewer #2: Yes

Reviewer #3: Yes

2. Has the statistical analysis been performed appropriately and rigorously? 

Reviewer #1: Yes

Reviewer #2: Yes

Reviewer #3: Yes

3. Have the authors made all data underlying the findings in their manuscript fully available?

Reviewer #1: Yes

Reviewer #2: Yes

Reviewer #3: Yes

4. Is the manuscript presented in an intelligible fashion and written in standard English?

Reviewer #1: Yes

Reviewer #2: Yes

Reviewer #3: Yes

5. Review Comments to the Author

Reviewer #1: This manuscript adopts the traditional SEIR model to simulate epidemic spread in Tianyi station at Ningbo city. The authors have conducted lots of numerical simulation, The paper is well organized and it is a interesting topic. The proposed materials and methods are described in detail afterwards. In general, I think that the article fits well into the scope of target journal. However, some revisions are required before the paper can be considered for publication.

1. Please give more explain for SEIR model, such as perturbation factor.

2. The discussion in the body of the manuscript is abundant and lack of logic, which makes it harded to understand. The authors need to reorganize the structure significantly.

3. Please give more explain for "1000 population" in your SEIR model.

Reviewer #2: The paper simulated the spread of communicable diseases in an urban rail transit station. Data were collected by a field investigation in the city of Ningbo, China. A SEIR-based model was developed to simulate the spread of infectious diseases in Tianyi station, considering four groups of passengers (susceptible, exposed, infected, and recovered) and a 14-day incubation period. Based on the historical data of infectious diseases, the parameters of the SEIR infectious disease model were clarified, and a sensitivity analysis of the parameters was also performed. The paper is clearly written and easy to understand. My comments are pointed out as follows:

1.To enrich your bibliography, some important references about SEIR infectious disease model should be added.

2. Please polish the quality of English language.

Reviewer #3: The article contains information technical and innovative that justifies its publication. The problem addressed is current and has technical relevance, which makes it significant. The paper is well organized and convincing. The simulation-based methodology is described as comprehensively. Interpretations and conclusions are justified by the results. The paper is well organized and convincing. My recommendations are：

1. The abstract can be rewritten to be more meaningful. The authors should add more details about their final results in the abstract.

2. Running a SEIR model for a 1,000 population without clear definition of the specifics of the problem is useless.

3. The calibration of the model parameter is not clearly described. Please check.

6. PLOS authors have the option to publish the peer review history of their article (what does this mean?). If published, this will include your full peer review and any attached files.

Reviewer #1: No

Reviewer #2: No

Reviewer #3: No

---

## [Author Response · Author response to Decision Letter 0]

24 Apr 2021

April 23, 2021

Dear Reviewers,

We would like to thank the reviewers for their constructive comments, which have significantly improved the quality of the paper. We have revised the manuscript according to your constructive suggestions, where the corresponding changes to your comments have been highlighted in RED in the revised version of the manuscript. The corresponding responses to your insightful comments are attached. 

Thank you again.

Corresponding author:

Name: Changxi Ma

E-mail: machangxi@mail.lzjtu.cn

---

## [Decision Letter · Decision Letter 1]

1 Jun 2021

Epidemic Spread Simulation in an Area with a High-density Crowd Using a SEIR-Based Model

PONE-D-21-06411R1

Dear Dr. Ma,

We’re pleased to inform you that your manuscript has been judged scientifically suitable for publication and will be formally accepted for publication once it meets all outstanding technical requirements.

Kind regards,

Yanyong Guo, Ph.D

Academic Editor

PLOS ONE

Additional Editor Comments (optional):

Reviewers' comments:

Reviewer's Responses to Questions

**Comments to the Author**

1. If the authors have adequately addressed your comments raised in a previous round of review and you feel that this manuscript is now acceptable for publication, you may indicate that here to bypass the “Comments to the Author” section, enter your conflict of interest statement in the “Confidential to Editor” section, and submit your "Accept" recommendation.

Reviewer #1: All comments have been addressed

Reviewer #2: All comments have been addressed

Reviewer #3: All comments have been addressed

2. Is the manuscript technically sound, and do the data support the conclusions?

Reviewer #1: Yes

Reviewer #2: Yes

Reviewer #3: Yes

3. Has the statistical analysis been performed appropriately and rigorously? 

Reviewer #1: Yes

Reviewer #2: Yes

Reviewer #3: Yes

4. Have the authors made all data underlying the findings in their manuscript fully available?

Reviewer #1: Yes

Reviewer #2: Yes

Reviewer #3: Yes

5. Is the manuscript presented in an intelligible fashion and written in standard English?

Reviewer #1: Yes

Reviewer #2: Yes

Reviewer #3: Yes

6. Review Comments to the Author

Reviewer #1: (No Response)

Reviewer #2: (No Response)

Reviewer #3: All my comments have been addressed by the authors in a well organized manner. The reviewer recommends the paper to be published.

7. PLOS authors have the option to publish the peer review history of their article (what does this mean?). If published, this will include your full peer review and any attached files.

Reviewer #1: No

Reviewer #2: No

Reviewer #3: No

---

## [Editor Report · Acceptance letter]

8 Jun 2021

PONE-D-21-06411R1 

Epidemic Spread Simulation in an Area with a High-density Crowd Using a SEIR-Based Model 

Dear Dr. Ma:

I'm pleased to inform you that your manuscript has been deemed suitable for publication in PLOS ONE. Congratulations! Your manuscript is now with our production department. 

Kind regards, 

on behalf of

Dr. Yanyong Guo 

Academic Editor

PLOS ONE